# The Clinical Characteristics of Immunoglobulin Light Chain Amyloidosis in the Chinese Population: A Systematic Scoping Review

Chengcheng Fu [1], Xiaohong Wang [2], Xian Cao [3], Lingjie Xu [3], Wang Liu [3], Jingnan Pi [3], Bin Wang [3] and Wenming Chen [4],*

1 The First Affiliated Hospital of SooChow University, 296 Shizi St, Cang Lang Qu, Suzhou 215006, China
2 Xi'an Janssen Pharmaceutical Ltd., 7F, Building One, Xinyan Mansion, No. 65, Guiqing Rd., Xuhui District, Shanghai 200233, China
3 Xi'an Janssen Pharmaceutical Ltd., 14F, Tower 3, China Central Place, No. 77, Jian Guo Road, Chaoyang District, Beijing 100012, China
4 Beijing Chaoyang Hospital, Capital Medical University, Gongren Tiyuchang Nanlu, Chaoyang District, Beijing 100020, China
* Correspondence: 13910107759@163.com

**Abstract:** Immunoglobulin light chain (AL) amyloidosis is the most common type of systemic amyloidosis in China and is associated with increased morbidity and a poor prognosis. However, the clinical characteristics of Chinese patients with AL amyloidosis have not been systematically investigated. This scoping review aimed to summarize the available literature regarding the clinical characteristics of patients with AL amyloidosis and identify potential knowledge gaps. We searched three electronic databases from inception to 7 February 2021. PICOS (Patient, Intervention, Comparison, Outcome and Study) design structure was used to formulate the data extraction. All statistical calculations and analyses were performed with R (version 3.6.0). Sixty-seven articles with 5022 patients were included. Results suggest Chinese patients were younger (57 years) at the time of diagnosis when compared with other patient populations and were predominantly male (61.2%). The time interval from the onset of symptoms to diagnosis was between 6 and 12 months. It was found that 41.1% of Chinese patients with AL amyloidosis were diagnosed with an advanced stage III disease when diagnosed, and 20.2% had a concurrent disease. The most involved organs were the kidneys (84.3%) and the heart (62.5%). In conclusion, our study shows some similarities and differences with other studies on the clinical characteristics of Chinese patients with AL amyloidosis, including the age at diagnosis, Mayo stage, and organ involvement. However, a nationwide epidemiological investigation is still needed to provide a comprehensive overview of this patient population in China.

**Keywords:** Immunoglobulin light chain; amyloidosis; clinical characteristics; Systematic Scoping Review





## 1. Introduction

Immunoglobulin light-chain (AL) amyloidosis is the most common type of systemic amyloidosis and is caused by the extracellular deposition of misfolded proteins as amyloid fibrils [1]. The abnormal protein deposition leads to protein toxicity and subsequently results in the loss of normal tissue structure as well as the dysfunction of corresponding organs, including the kidneys, heart, liver, and nervous system [2]. The presentation varies greatly depending on the target organ involved. The clinical manifestations of AL amyloidosis often overlap with other more prevalent chronic diseases [3]. Non-specific early symptoms of AL amyloidosis do not always attract the attention of patients and physicians doctors and may lead to delays in seeking medical advice and in reaching a diagnosis. Physician reports have noted the time interval from the onset of symptoms to the pathological confirmation of a diagnosis is approximately 10 months [3,4]. Because AL amyloidosis is a progressive disease, a late diagnosis is associated with a poor prognosis [5].

The incidence of AL amyloidosis is approximately 12 cases per million persons per year in the US and European countries [6]. The majority of patients with AL amyloidosis are elderly [7]. With the gradual aging of populations in most high- and middle-income countries, the incidence rate of AL amyloidosis has been increasing in recent years [5,8]. However, epidemiological studies of the incidence of AL amyloidosis and the clinical characteristics of patients in China remain sparse. Consequently, this paucity of data has led to insufficient recognition of the disease characteristics, creating obstacles for physicians in clinical practice.

Therefore, we initiated a series of scoping reviews to provide an overview of the epidemiology, clinical burden, diagnostic techniques, therapeutic regimens, and prognostic factors in Chinese patients with AL amyloidosis. Scoping reviews are used to present a landscape of existing heterogeneous studies whose topics are complex in nature, to identify evidence gaps, and to guide future research initiatives. The current scoping review is the first in a series that aims to describe the distribution of the studies, the prevalence of the disease, the types of AL amyloidosis, and the clinical characteristics of patients. The results will be used to (1) inform future research on AL amyloidosis, and (2) lay a foundation for generating comprehensive evidence for healthcare providers. The treatment, therapy, diagnosis, and prognosis of AL amyloidosis are not presented in this review and will be discussed in future manuscripts.

## 2. Materials and Methods

A scoping review is defined by Arksey and O'Malley "to map rapidly the key concepts underpinning a research area and the main sources and types of evidence available and can be undertaken as standalone projects in their own right, especially where an area is complex or has not been reviewed comprehensively before" [9]. The current scoping review was performed according to the preferred reporting items for a scoping review (PRISMA) extension statement [10], incorporating the methods introduced and advanced by Arksey et al. [11,12]. The protocol has been registered in the International Platform of Registered Systematic Review and Meta-analysis Protocols (INPLASY) with the identification number 'INPLASY202190086'.

### 2.1. Search Strategy

A systematic search in MEDLINE, EMBASE and CNKI databases was conducted by XC and XW without limitations on the publication date and document type. The following items were used to develop our search strategy: "amyloidosis", "amyloidos *", "amyloido *", "AL amyloidosis" and "China." The details of the search strategies are presented in Supplementary Materials (supplemental search strategies).

### 2.2. Eligibility Criteria and Literature Screening

Both interventional and non-interventional studies that reported Chinese patients with AL Amyloidosis between 1 January 2010, and 7 February 2021, were included in this study. Chinese patients with AL Amyloidosis were defined as study participants enrolled in healthcare institutes in mainland China. We excluded case reports, reviews, consensus, thesis, and questionnaires. We divided included studies into three categories, namely by period of recruitment, recruit setting, and diagnosis: "Different study population", "Partially overlapping study populations" and "Fully replicated study populations". "Different study populations" or "Partially overlapping study populations" were included for analysis of the clinical characteristics of the population. The project flowchart details the steps for determining duplicate studies (Figure S1). Two reviewers screened studies after reading the titles and abstracts obtained from the search results. All potentially relevant citations were requested and inspected in detail using the full-text version. Disagreements were resolved by discussion, with assistance from the senior author (CF) if necessary. A PRISMA flow diagram was constructed to show the full study-selection process (Figure 1).

*2.3. Data Extraction*

Data from each article were extracted by one reviewer and double-checked by another reviewer using a standardized data extraction form (Table S1). Discrepancies between authors were resolved by the senior author (CF). In cases where information from a potentially included study was lacking, the corresponding authors were contacted, and further information was requested. A PICOS (Patient, Intervention, Comparison, Outcome, Study) design was used to formulate the data extraction. Extracted information included (1) study characteristics, the first author's name, the publication year, trial registration number, hospital, province, center, study design, diagnostic tests, recruiting period and sample size; (2) the clinical characteristics of the patients include the following: age, gender, type of AL amyloidosis, Mayo stage, the proportion of secondary diseases, organ involvement, the difference between the involved and uninvolved light chain (dFLC), estimated glomerular filtration rate (eGFR), N-terminal pro-B-type natriuretic peptide (NT-proBNP), cardiac troponin T (cTnT), serum creatinine and 24 h urinary protein.

*2.4. Statistical Analyses*

All statistical calculations and analyses were performed with R (version 3.6.0) [13]. Categorical data were expressed as the number of events and the proportion and were pooled using the fixed-effect model. Continuous variables were expressed as mean and standard deviation (SD) or median (interquartile range, IQR) as reported in the original publications. To calculate the sample size at each site in multicenter studies, we assumed equal proportions of samples enrolled in each study site. We also drew a trajectory of the number of patients being diagnosed each year. For the calculation of the sample size being diagnosed each year, one study could potentially enroll patients over a wide range of years (for instance, from 2002 to 2009). In this situation, we assumed an equal monthly enrollment number to assign the samples to each year. The trajectory of the sample size was smoothed using the locally weighted regression (LOESS) method with the span set to 0.2 using the R package 'ggplot2'. When describing the characteristics of patients, we pooled the mean value of the characteristics (such as age) reported by each study. Where mean values were not reported, the median was used as a surrogate. Data were pooled using a fixed-effect model. For biomarkers, the proportion of patients with values lower than a specific threshold was calculated. Without assuming independent samples from the same normal distribution, the proportions of patients were calculated in each study using their respective mean and standard deviation values for applicative chemical indicators, then pooled using the fixed-effect model.

**3. Results**

*3.1. Description of the Included Studies*

A total of 770 articles were retrieved from the three databases: MEDLINE, EMBASE and CNKI. After preliminary screening and re-screening, 120 articles [14–133] were finally included (Figure 1, Table S2). The included articles are from 15 Chinese cities, of which Beijing had the highest proportion (42.5%), followed by Nanjing (19.2%) and Chengdu (8.3%) (Figure 2). The majority of the included articles were published in 2019 (20.0%), followed by 2020 (15.0%), and 2016 (13.3%). Of these 120 articles, 29 were interventions, 10 were diagnostic, 39 were prognostic, and 42 aimed to analyze clinical characteristics. Figure 3 shows the distribution of article types. In recent years, there has been an apparent increase in the number of published articles evaluating treatments and diagnostic procedures as well as a marked increase in prognostic articles.

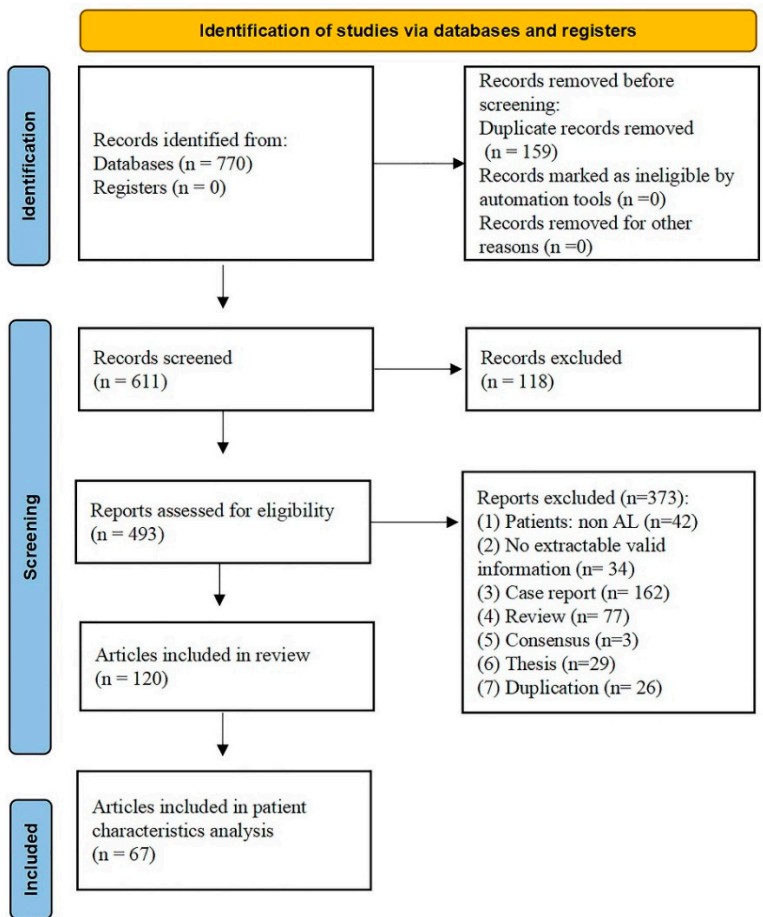

**Figure 1.** Literature screening flow chart.

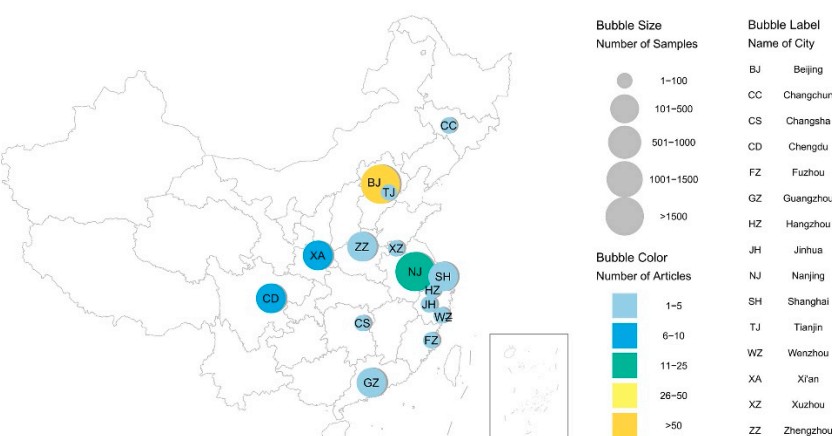

**Figure 2.** Map of regional distribution and number of articles.

After identifying and removing duplicate articles, 67 articles with a total of 5022 patients were finally included for analysis of baseline characteristics [15–21,24–26,31,32,34,35,42–45,50, 52,53,55–57,60,61,63–66,69–74,76,77,80,82,83,85,87–90,93,96,97,99–104,106,109–111,114,115,118, 120,122,124,131,133]. These patients came from 15 cities, the majority being from Nanjing (34.3%), Beijing (34.1%), and Chengdu (10.2%). The number of patients enrolled increased annually in the articles published between 2000 and 2015 and reached a peak in 2015. Table S3 shows the patient characteristics in the included articles.

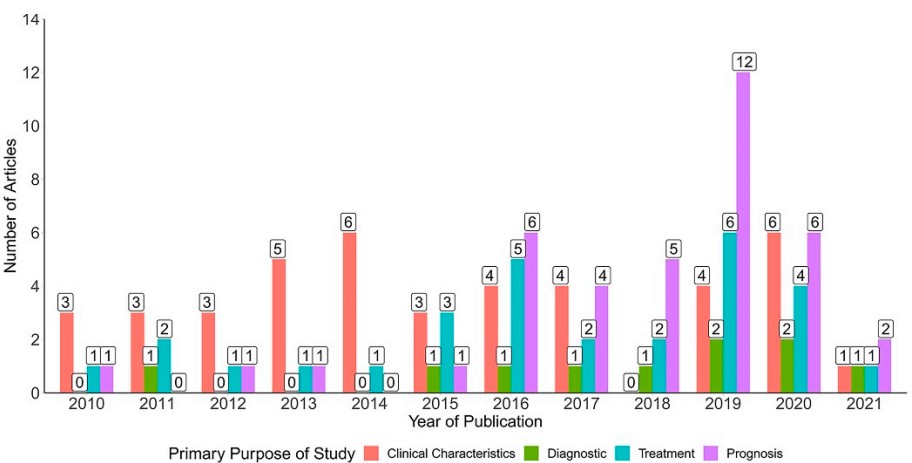

**Figure 3.** Year of publication and number of articles.

### 3.2. Clinical Characteristics of Patients with AL Amyloidosis

The mean age of patients with AL amyloidosis ranged from 25 to 85 years, and the calculated weighted average age was 57 years. The majority (61.2%) of patients were male. The mean BMI ranged from 21.7 to 24.3 kg/m$^2$, and the calculated weighted mean BMI was 22.8 ± 3.2 kg/m$^2$. The median time interval from the onset of symptoms to diagnosis was 6 to 12 months.

Table 1 shows the clinical characteristics of patients in the included articles. A total of 34 articles reported whether AL amyloidosis was with or without concurrent diseases, of which 22 articles included only patients with AL amyloidosis alone, two articles included only AL amyloidosis patients with concurrent multiple myeloma and another ten articles included patients with or without concurrent diseases. Among 698 patients included in those ten articles, 20.2% were with concurrent diseases. Nine articles provided information about the presence of concurrent diseases, which were multiple myeloma in the vast majority of cases (128/130, 98.5%). Forty-two articles including 3610 patients reported the patient's disease subtype, of which 80.0% were λ type. A total of 18 articles with 2020 patients reported newly diagnosed or refractory AL amyloidosis (whether the treatment plan was a first-line or non-first-line treatment), of which only 10 patients (0.5%) had refractory AL amyloidosis. Eleven articles with 1559 patients reported the Mayo 2004 staging, and 41.1% of the cases were stage III. Thirteen articles with 1585 patients reported the Mayo 2012 staging, of which over 40% were advanced stage (stage III–IV). Among 929 patients for whom the distribution of the New York Heart Association (NYHA) functional class was reported, approximately 1/3 were grade III–IV (Table 1).

**Table 1.** Summary of clinical characteristics of patients.

| Characteristics | Toal Number (n) | Positive Number (n) | Percentage (%) |
|---|---|---|---|
| Type of light chains | | | |
|     Lambda | | 2890 | 80.0 |
|     Kappa | 3610 | 508 | 14.1 |
|     Unclear * | | 212 | 5.9 |
| Disease course | | | |
|     Newly diagnosed | | 2010 | 99.5 |
|     Relapsed/Refractory | 2020 | 10 | 0.5 |
| Mayo 2004 cardiac staging system | | | |
|     I | | 383 | 24.6 |
|     II | 1559 | 535 | 34.3 |
|     III | | 641 | 41.1 |

**Table 1.** *Cont.*

| Characteristics | Toal Number (n) | Positive Number (n) | Percentage (%) |
|---|---|---|---|
| Mayo 2012 cardiac staging system | | | |
| I | | 532 | 33.6 |
| II | | 391 | 24.7 |
| III | 1585 | 399 | 25.2 |
| IV | | 263 | 16.6 |
| NYHA functional class | | | |
| I–II | | 638 | 68.7 |
| III–IV | 929 | 291 | 31.3 |
| Organ involvement | | | |
| Renal | 3798 | 3202 | 84.3 |
| Cardiac | 4070 | 2542 | 62.5 |
| Soft tissue | 807 | 335 | 41.5 |
| Gastrointestinal tract | 2174 | 747 | 34.4 |
| Liver | 3190 | 578 | 18.1 |
| Nerves | 2381 | 268 | 11.3 |
| Renal | 3798 | 3202 | 84.3 |
| Number of involved organs | | | |
| 1 | | 79 | 20.7 |
| 2 | | 122 | 31.9 |
| 3 | 382 | 121 | 31.7 |
| ≥4 | | 60 | 15.7 |
| dFLC (median, mg/L) | | | |
| 19.1–250.3 | 2044 | | |
| <50 | | 312 | 15.3 |
| ≥50 | | 1732 | 84.7 |
| <180 | | 1404 | 68.7 |
| ≥180 | | 640 | 31.3 |
| NT-proBNP (median, ng/L) | | | |
| 222–14,213 | 2928 | | |
| <332 | | 512 | 17.5 |
| ≥332 | | 2416 | 82.5 |
| <1800 | | 1392 | 47.5 |
| ≥1800 | | 1536 | 52.5 |
| TNT (median, µg/L) | | | |
| 0.016–0.21 | 1545 | | |
| <0.035 | | 1038 | 67.2 |
| ≥0.035 | | 507 | 32.8 |
| eGFR (median, mL/min/1.73 m$^2$) | | | |
| 70–115.3 | 1259 | | |
| <50 | | 140 | 18.4 |
| ≥50 | 764 [‡] | 624 | 81.6 |
| 24 h urine protein (median, g/24 h) | | | |
| 0.9–6.64 | 2423 | | |
| <5 | | 2140 | 83.3 |
| ≥5 | | 283 | 11.7 |

Abbreviations: AL amyloidosis, immunoglobulin light chain amyloidosis; NYHA, New York Heart Association Classification; dFLC, difference between the involved and uninvolved light chain; eGFR, estimated glomerular filtration rate; NT-proBNP, N-terminal pro-B-type natriuretic peptide; cTnT, cardiac troponin. *: If the article reported only one type of light chain (Lambda or Kappa), the remaining population was deemed to be unclear. [‡]: The proportion of patients with eGFR values less than and greater than 50 mL/min/1.73 m$^2$ was calculated from 12 articles (n = 764) that reported the mean and standard deviation of eGFR.

### 3.3. Organ Involvement

A total of 41 articles with 4168 patients reported organ involvement. The most common organ affected was the kidney, followed by the heart, skin and soft tissue (Table 1). Four articles reported the percentage of patients (n = 382) with a different number of involved organs, 79.3% of patients involved more than one organ (Table 1).

*3.4. Proportion of Plasma Cells in Bone Marrow*

Of 1241 patients in 11 articles, the proportion of bone marrow plasma cells was ≥10 in 85 patients (6.8%). The median range (n = 922) was 1.5–4.5%.

*3.5. Prognostic Indicators*

We also compared the distribution of baseline characteristics with previously validated prognostic threshold values (Table 1). The median value of dFLC ranged from 19.1 to 250.3 mg/L. As for cardiac prognostic markers, the median values of NT-ProBNP and cTnT ranged from 222 to 14,213 ng/L and from 0.016 to 0.21 µg/L, respectively. As for renal prognostic markers, the median values of eGFR and 24 h urine protein ranged from 70 to 115.3 mL/min/1.73 m$^2$ and from 0.9 to 6.64 g/24 h, respectively. An analysis of 12 articles including 764 patients revealed the proportion of patients with an eGFR value <50 mL/min/1.73 m$^2$ was 18.4% (ranging from 10.0% to 32.6%).

**4. Discussion**

*4.1. Summary of Evidence*

To the best of our knowledge, this is the first study investigating the clinical characteristics of Chinese patients with AL amyloidosis based on 67 published articles with 5022 Chinese patients with AL amyloidosis. We observed that found articles were concentrated in the first- and second-tier cities and that recent years have seen a slight increase in the number of articles. The findings from this review offer insights into the clinical characteristics of AL amyloidosis in this patient population, including but not limited to age at diagnosis, gender, the time interval from onset of symptoms to diagnosis, disease stage, organ involvement, as well as values of blood and urinary markers (Table S1 and Table 1).

*4.2. Comparison with Other Studies*

The age distribution of Chinese patients with AL amyloidosis is relatively broad, ranging from 25 to 85 years, with a mean of 57 years, while the previous mean age of patients with AL amyloidosis in North America and Italy has been reported to be of was 63 and 70 years, respectively [1,134]. A nationwide or a regional epidemiological study is therefore warranted to better assess the demographic characteristics of this patient population in China. We found the sex distribution, organ involvement and light chain type was similar to those characteristics reported in studies from Japan, Korea and the US [135–137].

As with other studies, AL amyloidosis is often underdiagnosed or misdiagnosed in China, and the time interval from the onset of symptoms to diagnosis was 6 to 12 months [4,138]. In a study by McCausland et al., the timeframe between symptom onset and the receipt of a diagnosis was 10 months (range 1 month to 2 years) [4]. In a survey conducted by Lousada et al., 63% of patients received an AL amyloidosis diagnosis within 1 year after the onset of symptoms [138]. Disease awareness among healthcare providers and how patients described their initial discomfort and whether sought early medical service is critical for the timely diagnosis of this disease [4]. The delayed diagnosis results in advanced multiorgan dysfunction (particularly cardiac in nature) and a poorer prognosis [4]. In our review, 79.3% of patients had more than one organ involved. While the heart is the most commonly affected organ in other populations [139], the kidney represented the majority (84.3%) of organ involvement in the included patient population.

Consistent with results from another report [140], the proportion of patients with stages III and IV disease (Mayo 2012 staging) was 25.2% and 16.6%, respectively. Some studies show that Mayo staging is an independent risk factor for patient survival [15,44]; thus, early diagnosis is needed so that care for this group can be improved. Fortunately, the number of patients receiving a timely and proper diagnosis of AL amyloidosis has increased in recent years as more attention and resources are being allocated to this clinical area [141].

According to our findings, the most involved organ seen in clinical practice is the kidney in Chinese patients with AL amyloidosis, while the heart is the primary target in patients from western countries [1]. This result may suggest that Chinese patients with AL amyloidosis have a greater susceptibility to renal involvement. An alternative explanation is that a diagnosis of kidney disease might have been more likely since most articles on Chinese patients with AL amyloidosis were evaluated in the hospitals' renal.

In the present study, due to the limitation of the reported value of biomarkers (only median or mean values were reported) of AL amyloidosis, we found that there was an obvious heterogeneity of patients among real-world studies. Therefore, further well-designed, multicenter, large-sample, prospective study is warranted to elucidate these concerns.

*4.3. Strengths and Limitations*

This review has certain strengths, this study had good quality control as we conducted the review strictly according to the Cochrane and PRISMA standard. We have provided an overview of Chinese studies evaluating AL amyloidosis. The findings from our review suggest directions for evaluating our summary of clinical characteristics of this patient population should help clinicians and investigators increase their awareness of how this disease presents.

There were also a few limitations that need to be considered when interpreting the findings. First, the scoping review only included data from published articles. This raises concerns about publication bias and the generalizability of our findings, as the characteristics of the population in published studies may not truly reflect those of the real-world population. Therefore, nationwide epidemiological studies are needed to draw an unbiased profile for patients with AL amyloidosis. Second, there may be partially duplicated samples in the literature even though we removed duplicate samples that could be identified based on the study's institutions, enrolled period time and authors.

## 5. Conclusions

This scoping review integrated the findings of current research on AL amyloidosis in China. The summary of clinical characteristics could be used as a general reference for disease insight and as a compass for future research. It may also provide healthcare providers with an overview of the characteristics of Chinese patients. As such, priority must be given to assessing each patient's clinical condition when referring to the findings from this review. However, the research on AL amyloidosis in China is still in its infancy and more multi-center, large sample, randomized controlled articles and multi-point epidemiological articles are needed to further provide higher-level evidence.

**Supplementary Materials:** The following supporting information can be downloaded at https://www.mdpi.com/article/10.3390/hemato4010002/s1. Search strategies; Figure S1 The project flowchart; Table S1 Standardized data extraction form; Table S2 Summary of included articles; Table S3 Summary of patient baseline characteristics in the included articles.

**Author Contributions:** Search strategy, X.C. and X.W.; draft protocol W.C. and B.W.; study selection, X.C. and X.W.; data extraction L.X., J.P. and W.L.; data analysis, C.F. and X.W.; writing review and editing, W.C. and C.F. All authors have read and agreed to the published version of the manuscript.

**Funding:** This research received no external funding.

**Institutional Review Board Statement:** Not applicable.

**Informed Consent Statement:** Not applicable.

**Data Availability Statement:** Not applicable.

**Acknowledgments:** This work was supported by Xi'an Janssen Pharmaceutical Ltd. We thank Xin Gao for his editorial assistance and Margueritte White for language editing and all other contributors who provided assistance and support but have not been mentioned. The collection and assembly

of data and statistical expertise were provided by Yang Zhang and Sai Zhao, from Systematic Review Solutions, Ltd.

**Conflicts of Interest:** Authors X.W., X.C., L.X., W.L., J.P. and B.W. were employed by Xi'an Janssen Pharmaceutical Ltd. during the study process. The remaining authors declared that the research was conducted in the absence of any commercial or financial relationships that could be construed as a potential conflict of interest. The authors are responsible for all content and editorial decisions and received no honoraria related to the development of this publication.

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
