# Peer review of "The Clinical Characteristics of Immunoglobulin Light Chain Amyloidosis in the Chinese Population: A Systematic Scoping Review"

_hemato, doi:10.3390/hemato4010002_

Round 1

Reviewer 1 Report

A major problem with such reports is the duplication of data; it is common and reasonable to use datasets that are expanding over time and published more than once.   There are 67 articles form 15 cities but given the rarity of the disease, how did the authors excluded studies that were repeating the inclusion of the same populations? Also, it is notable that reports from 3 cities comprised ~80% of reports.

The tables are somewhat confusing. I am not sure what they show: do they present the characteristics or the numbers of studies or something else?. For example , table 3 is entitled “Summary of clinical characteristics of patients” and the data presented with percentages are not what one would expect according to the paragraph referencing the table.  It is probably the structure of the table , so please restructure. Table 4 is more informative and clear and perhaps table 3 should also follow the structure of table 4. Same for table 5   

Discussion: “Age distribution of Chinese patients with AL Amyloidosis is relatively broad, ranging from 25 to 85 years and the mean age was 57 years. However, studies based on populations from north America and Japan showed that the median age of patients were around 65 years”: first the age distribution is what one would expect, second the authors compare mean to median age and this may not be the correct approach. They should probably compare median or means unless the distribution is gaussian (which is not in these diseases).

I don’t think that such a long paragraph on strength and limitations is needed

Overall I see mostly similarities that differences with other studies from US or Europe. What is not reported however is the treatments given and the outcomes  

Author Response

Dear editors and reviewers:

Thank you for your letter and for the reviewers’ comments of our manuscript entitled " The Clinical Characteristics of Immunoglobulin Light Chain Amyloidosis in the Chinese Population: A Systematic Scoping Review") (ID: hemato-1934517). We modified the information as the reviewers pointed. Please see the response to the reviewers' comments point to point:

Point 1: A major problem with such reports is the duplication of data; it is common and reasonable to use datasets that are expanding over time and published more than once. There are 67 articles form 15 cities but given the rarity of the disease, how did the authors excluded studies that were repeating the inclusion of the same populations? Also, it is notable that reports from 3 cities comprised ~80% of reports.

Response: Thanks for reviewers’ concerns. Studies that were repeating the inclusion of the same populations were excluded according to the process in Fig S1. We divided 120 studies into three categories, by time of recruitment, location of recruitment and diagnosis. They are "Different study population", "Partially overlapping study population" and "Full replicated study population". "Different study population" or “Partially overlapping study population” were included for analysis. In the article, we added a detailed description.

When studies were recruited in different location or at different times, we consider them to be “Different study population”. When the recruitment sites were the same between studies, and the time overlap was less than 6 months, which also be considered as “Different study population”. "Full replicated study population" included studies where the recruitment time, location, and diagnosis were completely replicated. Also, when the diagnostic scope of one study was covered by another study, that study was considered as "Full replicated study population". When the locations of recruitment were the same between studies, and the time overlap was greater than 6 months, which be identified as "Partially overlapping study population". "Partially overlapping study population" But we can't sure from the articles whether there are overlapping populations.

According to several literature reports, the incidence of AL amyloidosis ranged from 2.4 to 8.9 per million person [1-4]. In this review, we included 5022 patients. Based on China's population of 1.4 billion, the number of patients included in this scoping review is within a reasonable range. Also, we noted that reports from Nanjing, Beijing and Chengdu comprised ~80% of reports. This phenomenon is justified because these three cities have a high level of medical care and they have the right hospitals and specialists for AL amyloidosis.

Point 2: The tables are somewhat confusing. I am not sure what they show: do they present the characteristics or the numbers of studies or something else? For example, table 3 is entitled “Summary of clinical characteristics of patients” and the data presented with percentages are not what one would expect according to the paragraph

referencing the table. It is probably the structure of the table, so please restructure. Table 4 is more informative and clear and perhaps table 3 should also follow the structure of table 4. Same for table 5.

Response: Thank you very much for the comments. We have restructured Table 3 and 5. Besides, we split the number of patients and percentage into two columns, we believe this presentation will make it easier for the reader to understand.

Point 3: Discussion: “Age distribution of Chinese patients with AL Amyloidosis is relatively broad, ranging from 25 to 85 years and the mean age was 57 years. However, studies based on populations from north America and Japan showed that the median age of patients were around 65 years”: first the age distribution is what one would expect, second the authors compare mean to median age and this may not be the correct approach. They should probably compare median or means unless the distribution is gaussian (which is not in these diseases). I don’t think that such a long paragraph on strength and limitations is needed.

Response: Thank you very much for the comments. We revised this content using mean age from two studies. Please check it in the discussion section.

Point 4: Overall I see mostly similarities that differences with other studies from US or Europe. What is not reported however is the treatments given and the outcomes.

Response: This suggestion is appreciated. The current scoping review is the first of the series to describe the distribution of the studies, the prevalence, types of AL amyloidosis, and clinical features of patients. Therapy, diagnosis, and prognosis of AL amyloidosis are not discussed in this scoping review, only because it is planned to include future paper. We redescribed this problem in the “Introduction” section.

Reference:

  1. Pinney JH, Smith CJ, Taube JB, et al. Systemic amyloidosis in England: an epidemiological study. Br J Haematol 2013;161:525-32.
  2. Hemminki K, Li X, Forsti A, et al. Incidence and survival in nonhereditary amyloidosis in Sweden. BMC Public Health 2012;12:974.
  3. Magy-Bertrand N, Dupond JL, Mauny F, et al. Incidence of amyloidosis over 3 ears: the AMYPRO study. Clin Exp Rheumatol 2008;26:1074-8.
  4. Bergesio F, Ciciani AM, Santostefano M, et al. Renal involvement in systemic amyloidosis – an Italian retrospective study on epidemiological and clinical data at Nephrol Dial Transplant 2007;22:1608-18.

Reviewer 2 Report

In this article, the authors systematically reviewed papers on AL amyloidosis in China. This work is well designed, and is significant regarding understanding of this rare disease characteristics in different areas.

Major criticism:
This thorough report does not include any data regarding therapy, outcomes and prognosis, but only data on clinical characteristics. It is implied in line 54 that this is the first out of more papers that are planned. If this is the reason, it should just be much better clarified. It should be emphasized that therapy, outcomes and prognosis are not discussed at all, only because it is planned to include those issues in a future paper, and the current paper only aims to discuss clinical characteristics.

Minor criticism:
• Line 38- "AL amyloidosis is often confused with other senile diseases" – AL is not a senile disease!
• Line 36- the peri-orbital ecchymosis, macroglossia, and the specific type of cardiac involvement are actually quite specific to Al
• Line 39- it should be better explained why late diagnosis is partially explains the poor prognosis. Some indolent diseases are also diagnosed after about 10 months, and does not necessarily confer poor prognosis...
• Table 2 is not needed. This data can be included in the text only.
• Table 3- all the data in the table is included in the text. Rather, the text should only highlight few issues and not duplicate the table.
• It should be considered to incorporate the data from table 4 and 5 into table 3 (to create one table only on patient characteristics).
• Line 200- the range of the renal markers is far less important than the median, it should be emphasized in the text and table 5.
• Line 239- this whole paragraph is not well explained and phrased.
• Line 247- this whole paragraph should also be rephrased

Author Response

Dear editors and reviewers:

Thank you for your letter and for the reviewers’ comments of our manuscript entitled " The Clinical Characteristics of Immunoglobulin Light Chain Amyloidosis in the Chinese Population: A Systematic Scoping Review") (ID: hemato-1934517). We modified the information as the reviewers pointed. Please see the response to the reviewers' comments point to point:

In this article, the authors systematically reviewed papers on AL amyloidosis in China. This work is well designed, and is significant regarding understanding of this rare disease characteristics in different areas.

Point 1: Line 38- "AL amyloidosis is often confused with other senile diseases" – AL is not a senile disease!

Response: This suggestion is appreciated. We revised this in Line 39-40 as follows: “The clinical manifestations of AL amyloidosis often overlap with other more prevalent chronic diseases.”

Point 2: Line 36- the peri-orbital ecchymosis, macroglossia, and the specific type of cardiac involvement are actually quite specific to Al

Response: Thank you very much for the comments. We have removed the relevant description from the manuscript.

Point 3: Line 39- it should be better explained why late diagnosis is partially explains the poor prognosis. Some indolent diseases are also diagnosed after about 10 months, and does not necessarily confer poor prognosis.

Response: Thank you very much for the comments. We revised this in Line 43-46 as follows: “Physician reports have noted the time interval from the onset of symptoms to the pathological confirmation of a diagnosis is approximately 10 months. Because, AL amyloidosis is a progressing disease, a late diagnosis is associated with a poor prognosis.” [1]

Point 4: Table 2 is not needed. This data can be included in the text only.

Response: The authors appreciated the suggestions. Since the data have been described in the text, we agreed to deleted Table 2.

Point 5: Table 3- all the data in the table is included in the text. Rather, the text should only highlight few issues and not duplicate the table.

Response: Thank you very much for the comments. We revised the text in the “Characteristics of patients with AL amyloidosis” section to highlight the characteristics of AL in Line 169-184, while the more specific data were in the table. Details are as follows: Table 2 shows the clinical characteristics of patients in the included articles. A total of 34 articles reported whether the AL amyloidosis was with or without concurrent diseases, of which 22 articles included only patients with AL amyloidosis alone, two articles included only AL amyloidosis patients with concurrent multiple myeloma, an-other ten articles included patients with or without concurrent diseases. Among of 698 patients included in those ten articles, 20.2% were with concurrent diseases. Nine arti-cles provided information about the presence of concurrent diseases, which was mul-tiple myeloma in the vast majority of cases (128/130, 98.5%). Forty-two articles including 3,610 patients reported the patient's disease subtype, of which 80.0% were λ type. A total of 18 articles with 2,020 patients reported newly diagnosed or refractory AL amyloidosis (whether the treatment plan was first-line or non-first-line treatment), of which only 10 patients (0.5%) had refractory AL amyloidosis. Eleven articles with 1,559 patients re-ported the Mayo 2004 staging, and nearly half the cases were stage III (41.1%). Thirteen articles with 1,585 patients reported the Mayo 2012 staging, of which over 40% were advanced stage (stage III-IV). Among 929 patients for whom the distribution of the New York Heart Association (NYHA) functional class was reported, approximately 1/3 were grade III-IV (Table 2).”

Point 6: It should be considered to incorporate the data from table4 and 5 into table 3 (to create one table only on patient characteristics).

Response: Thank you very much for the comments. As Table3, 4 and 5 correspond to different paragraphs and content respectively (table 3 summarized clinical characteristics of patients, table 4 summarized organ involvement in patients and table 5 summarized laboratory biomarkers for evaluating the severity and pattern of organ involvement), we think that table 4 and 5 should not be combined with table 3 into a single table. Presenting them separately could make them clearer to the reader.

Point 7: Line 200- the range of the renal markers is far less important than the median, it should be emphasized in the text and table 5.

Response: The authors appreciated the suggestions. Due to the limitations of the data source, we were unable to obtain the raw data and could only extract the median value from each of the original studies. Thus, in Line 205-207, we described range of median value of eGFR, not range of eGFR.

Point 8: Line 239- this whole paragraph is not well explained and phrased.

Response: Thank you very much for the comments. We revised this paragraph in “Comparison with other studies” section in Line 234-245 as follows: “Consistent with other studies, the time interval from the onset of symptoms to diagnosis was 6 to 12 months. In a study by McCausland et al, the timeframe between symptom inset and the receipt of a diagnosis was 10 months (range 1 month to 2 years). In a survey conducted by Lousada et al, 63% patients received AL diagnosis of within 1 year after the onset of symptoms. In clinical settings, delayed diagnosis of AL amyloidosis is often under-mis-diagnosed depend on a among even develop of presumed unknown etiology. How patients described their initial discomfort and whether sought early medical service. The delayed diagnosis results in advanced multiorgan dysfunction (particularly cardiac in nature) and a poorer prognosis. In our review, 79.3% of patients had more than one organ involved. While the heart is the most commonly affected organ in other populations, the kidney represented the majority (84.3%) of organ involvement in our patient population.”

Point 9: Line 247- this whole paragraph should also be rephrased.

Response: The authors appreciated the suggestions. We modified this paragraph in “Comparison with other studies” section Line 246-251 as follows: Consistent with results from another report [140], the proportion of patients with stages III and IV disease (Mayo 2012 staging) was 25.2% and 16.6%, respectively. Some studies show that Mayo staging is an independent risk factor for patient survival, thus, early diagnosis is needed so that care for this group can be improved. Fortunately, the number of patients receiving a timely and proper diagnosis of AL amyloidosis has in-creased in recent years as more attention and resources are being allocated to this clinical area.”

Reference:

  1. Schulman A, Connors LH, Weinberg J, et al. Patient outcomes in light chain (AL) amyloidosis: The clock is ticking from symptoms to diagnosis. Eur J Haematol. 2020;105:495-501.

Round 2

Reviewer 1 Report

No additional comments 

Author Response

Dear editors and reviewers:

Thank you for your letter and for the reviewers’ comments of our manuscript entitled " The Clinical Characteristics of Immunoglobulin Light Chain Amyloidosis in the Chinese Population: A Systematic Scoping Review") (ID: hemato-1934517). We modified the information as the reviewers pointed. Please see the response to the reviewers' comments point to point:

Point 1: Moderate English language and style required.

Response: Thank you for your recommendation. We have asked a native speaker to modify and rephrase the manuscript, and the current language and style should be fine.

Reviewer 2 Report

-          Line 49- progressing should be corrected to progressive

-          One really needs to struggle in order to understand table 1 and 2, as its' data and titles are quite similar. Is table 1 meant only to show how many articles reported on a specific parameter, for example, troponin? If so, this is not clear at all, and not crucial, and hence should be included only as a supplemental table. The most important table is table 2, which summarize the patients' characteristics.  However, it is not reporting on important variables that are in table one- time to diagnosis, organ involved and more. Organ involvement is included in a separate table, and this data can be merged into table 2. Same regarding table 4.

-          In table 4 (that should be merged to table 2), eGFR is presented weirdly with only one option (70-115.3), no No. of pts and proportion.

Author Response

Dear editors and reviewers:

Thank you for your letter and for the reviewers’ comments of our manuscript entitled " The Clinical Characteristics of Immunoglobulin Light Chain Amyloidosis in the Chinese Population: A Systematic Scoping Review") (ID: hemato-1934517). We modified the information as the reviewers pointed. Please see the response to the reviewers' comments point to point:

Point 1: Line 49- progressing should be corrected to progressive.

Response: This suggestion is appreciated. We revised this in Line 44 as follows: “Because, AL amyloidosis is a progressive disease, a late diagnosis is associated with a poor prognosis.”

Point 2: One really needs to struggle in order to understand table 1 and 2, as its' data and titles are quite similar. Is table 1 meant only to show how many articles reported on a specific parameter, for example, troponin? If so, this is not clear at all, and not crucial, and hence should be included only as a supplemental table. The most important table is table 2, which summarize the patients' characteristics.  However, it is not reporting on important variables that are in table one- time to diagnosis, organ involved and more. Organ involvement is included in a separate table, and this data can be merged into table 2. Same regarding table 4.

Response: Thank you for your suggestion. We moved Table 1 to supplementary materials as supplemental Table S3. Furthermore, we combined the original Table 2, Table 3, Table 4 into one large table as Table 1: Summary of clinical characteristics of patients.

Point 3:  In table 4 (that should be merged to table 2), eGFR is presented weirdly with only one option (70-115.3), no No. of pts and proportion.

Response: Thank you for your suggestion. For biomarker eGFR, instead of calculating the proportion of patients according to different cut off values, we calculated the proportion of patients with an eGFR value < 50 mL/min/1.73m2 according to the articles that reported the mean eGFR value. The number and proportion of patients have been added in Table 1. Specific methods are presented in the statistical analyses section in Line 125-129: For biomarkers, the proportion of patients with values lower than a specific threshold was calculated. Without assuming independent samples from the same normal distribution, proportions of patients were calculated in each study using their respective mean and standard deviation values for applicative chemical indicators, then pooled using the fixed-effect model. The detailed description was given in Line 198-200: Analysis of 12 articles including 764 patients revealed the proportion of patients with an eGFR value < 50 mL/min/1.73m2 was 18.4% (ranging from 10.0% to 32.6%).

Round 3

Reviewer 2 Report

I reviewed again the paper and recommending to accept in its current form without any further comments (except for another text editing for typos from your end, if possible).